# Decoder-Only LLMs are Better Controllers for Diffusion Models

| Dalle-3 | Ours | Dalle-3 | Ours |
|---|---|---|---|

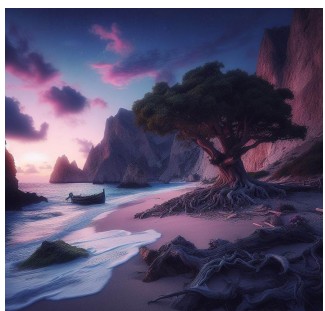 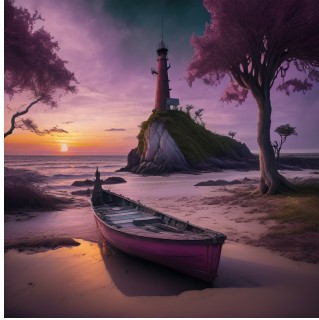 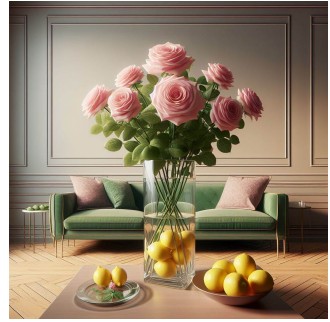 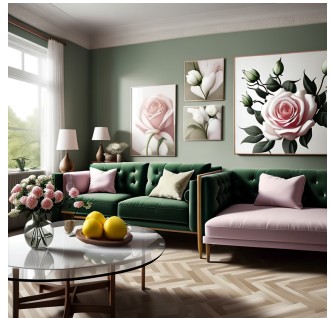

**Text Prompt:** Envision a secluded beach at twilight, where the last golden rays of the sun cast a soft glow on the rugged cliffs. The sky is a tapestry of pink and purple, with the early evening stars just beginning to emerge. Waves gently lap against the shore, where a lone, ancient oak tree stands, its twisted roots spilling out onto the white sand. Nearby, a weathered boat rests in the sand, abandoned, its paint peeling and sails tattered. In the background, a lighthouse perches precariously at the cliff's edge, its beacon flickering to life as the light fades. The scene is one of serene isolation, untouched by the rush of modern life, a timeless testament to nature's quiet majesty.

**Text Prompt:** In the room, there is a transparent glass vase filled with beautiful pink roses. Three lemons are placed next to the vase on a table. The table is a next to a green couch. The couch is adorned with two pillows, One is pink and the other has a green and gray pattern. The floor is covered with wooden flooring. The walls are painted with latex paint. The ceiling is made of plaster and has a line-shaped design. The room has a large window, providing ample natural light.

**Figure 1: Comparison of our LLMDiff with DALL-E 3 [2]. The pink texts represents the parts that our model has understood but DALL-E 3 has not. Diffusion Models like DALL-E 3 that are based on text encoders are prone to neglecting details when interpreting long complex texts, and they lack a comprehensive understanding of entity relationships. Instead, our model, which employs a decoder-only LLM, can more effectively grasp semantic and logical relationships between entities. As a result, it generates images that more accurately align with the user's intent.**

## ABSTRACT

Groundbreaking advancements in text-to-image generation have recently been achieved with the emergence of diffusion models. These models exhibit a remarkable ability to generate highly artistic and intricately detailed images based on textual prompts. However, obtaining desired generation outcomes often necessitates repetitive trials of manipulating text prompts just like casting spells on a magic mirror, and the reason behind that is the limited capability of semantic understanding inherent in current image generation models. Specifically, existing diffusion models encode the input text prompt with a pre-trained encoder structure, which is usually trained on a limited amount of image-caption pairs. State-of-the-art large language models (LLMs) based on the decoder-only structure have shown very powerful semantic understanding capability as their architectures are more suitable for training on very large-scale unlabeled data. In this work, we propose to enhance text-to-image diffusion models by borrowing the strength of semantic understanding from large language models (LLMs), resulting in a simple yet effective adapter to allow the diffusion models to be compatible with the decoder-only structure. In the evaluation, we conduct not only extensive empirical results but also the supporting theoretical analysis with various architectures (e.g., encoder-only, encoder-decoder, and decoder-only). The experimental results show that the enhanced models with our adapter module are superior to the stat-of-the-art models in terms of text-to-image generation quality and reliability.

## 1 INTRODUCTION

Image generative models have progressed explosively in recent years, with the prevalence of Generative Adversarial Networks (GANs) and diffusion models. Text-to-image generation methods such as Stable Diffusion [24, 27], DALL-E 3 [2], and Imagen [28] are capable of synthesizing high-quality images by taking textual descriptions (prompts) as the input. One key step of these models is to understand the user intention and semantic meanings from the text prompts and encode them to text features for further driving image content generation with diffusion models. To this end, most of the existing methods adopt an encoder-based language model structure (*e.g.*, CLIP [25] or T5 [26]), which were pre-trained on limited amount of image-caption pairs or texts pairs due to the expensive data annotation cost, resulted in the unsatisfying performance for the image generation quality and reliability. As a result, obtaining a user-desired image with these methods is very hard especially with a purpose of generating a complex and detail-rich image, and repetitive trials on manipulating the text prompts are nearly a must have. For example, as shown in Fig. 1, the state-of-the-art DALL-E

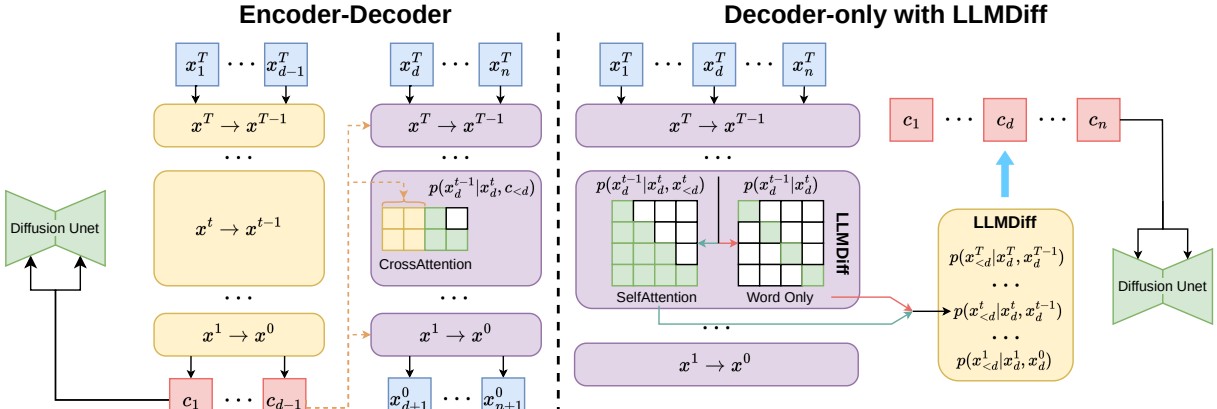

**Figure 2: Comparison with other neural network structures employed for computing text encoding in diffusion models. Our proposed LLMDiff, which leverages a decoder-only structure with casting the transformer-based language model as a diffusion model, can predict the text encodings for text-to-image generation by integrating layer-wise representations in the language model. Intuitively, compared with other structures (*e.g.* encoder-decoder) our LLMDiff is more powerful on exploring the semantic meanings and dependency among words from the input text prompt. Please refer more details and theoretical derivations in Sec. 3.4.**

3 fails to comprehend the entities and their relationships described in the complex prompts, resulting in numerous omissions.

On the other hand, we have also witnessed a very fast development on the Large Language Models (LLMs), *e.g.*, GPT-4 [22], PaLM [7] and Llama2 [14], which have show very incredible power on semantic understanding, reasoning and naturally interacting with human. These LLMs mainly employ the decoder-only structure that can be trained on a massive scale of unlabeled textual data. Unfortunately, bridging the ability of LLMs with current diffusion-based text-to-image generation framework is unexplored due to the incompatibility of these two model architectures. Some recent attempts have made on borrowing the ability of LLMs for enhancing the text-to-image generation performance with the diffusion models [11, 18]. Their approaches proposed to enrich or rewrite the user text prompt through LLMs and still rely on the vanilla text encoders to guide the image generation process within the diffusion models, leading to sub-optimal performance.

To tackle this challenge, we propose a novel and general approach to upgrading various text-to-image diffusion models by borrowing the strength of semantic understanding from large language models (LLMs). In particular, we reveal that a Transformer-based language model (*e.g.* ChatGPT [22]) can be rephrased as the denoising steps in Denoising Diffusion Probabilistic Models (DDPMs) [12]. Viewing LLMs as diffusion models, we have further derived theoretical underpinnings for extracting text encodings from the blocks of LLMs. These findings drive us to attach a simple yet effective network module to the cross-attention part of the denoising U-Net. as shown in Fig. 2. And this module enable us to effectively integrate block-wise representations within the language model for generating the text encoding of the input text prompt, which can accurately capture the semantic meanings and contextual dependency among words due to the power of pre-trained LLMs. We name this module as LLMDiff-Adapter as it can be very compatible

plug and play component for connecting LLMs with various text-to-image diffusion models and gaining conspicuous improvement. As some examples shown in Fig. 1, the results generated by our model can better preserve the semantic meanings and user intent from the input prompts, *e.g.* well representing the entities and their relationships for the image generation.

In the evaluation, we conduct a comparative analysis on different text-to-image models on the same benchmarks. We compared the performance of using our proposed LLMDiff-Adapter against other architectures, *e.g.* simply connecting the output of decoder-only LLMs or adopting encoder structure such as CLIP [25] and T5 [26] through linear layers to the diffusion models. The experimental results show that our model achieve superior performance among the competitions in several aspects, including the quality of generated image details, logical coherence, and comprehensive understanding of the text descriptions. The relevant quantitative results are also presented for underscoring the effectiveness of our approach in solving the limitations of current diffusion-based text-to-image generation methods.

## 2 RELATED WORKS

### 2.1 Text-to-Image Diffusion Models

Recently, diffusion-based image generation models have achieved remarkable success. These models learn to iteratively denoise a noisy image and generate the image progressively [12]. Compared to GAN-based methods, diffusion models are more stable in training and able to generate more diverse images. With the advent of diffusion models incorporating guidance mechanisms [13, 19], there has been a notable advancement in the performance of diffusion models. For the first time, diffusion models beat GANs in conditional generation tasks. Ever since, the focus of research on text-to-image synthesis has gradually shifted from GAN to Diffusion [5, 15, 16, 21]. Some large-scale text-to-image models [2, 10, 27, 31] have achieved highly accurate and fine-grained controllable semantic generation.

The recently proposed latent diffusion model (LDM) [27] unprecedentedly makes high-resolution and high-quality text-to-image models become a reality. Based on LDM, DALL-E 3 [2] has ushered the text-to-image models into unprecedented levels, leveraging powerful text encoders and high-quality data.

## 2.2 Large Language Models

In recent years, large-scale language models with billions of parameters have demonstrated remarkable performance across various natural language understanding and generation tasks. The dominant form of language models shifted from BERT-like models [9, 20] that focus on language understanding to the currently prevalent generative language models with decoder-only architectures [7, 14, 17, 22]. These decoder-only models have successfully unified a wide spectrum of tasks, showing commendable proficiency in dialogue interactions. Even in language comprehension tasks, [4, 8] also shows that CLIP and BERT style text encoders perform worse than decoder-only LLMs. Moreover, recent models exhibit the ability of in-context learning [3], enabling them to adaptively leverage contextual information to accomplish downstream tasks.

## 2.3 LLMs for Text-to-Image Generation

Existing text-to-image diffusion models are primarily based on encoder-structured text models like CLIP and T5. However, there are ongoing efforts of works that seek to explore the potential for transposing the wealth of knowledge inherent in Large Language Models (LLMs) into existing diffusion frameworks. Certain research endeavors, such as [11, 18], have attempted to utilize LLMs to predict the layout of objects, thereby enhancing the logical coherence and overall quality of the images produced by diffusion models. This is achieved by employing LLMs to rewrite the prompts, ensuring a better alignment between the generated images and the input text. Another approach [1] attempts to use LLMs to help users construct better prompts, leveraging the capabilities of LLMs to generate superior images.

While these pioneering efforts are indeed instrumental in integrating the knowledge of LLMs into diffusion models, they predominantly employ indirect methods to bridge the gap between them, and thus, are inherently constrained by the limitations of the inefficient text encoder. In contrast, we propose a novel method that directly integrates the output of the LLM into the existing diffusion model. By completely discarding the text encoder, we aim to liberate the text-to-image diffusion models from the bottleneck of language comprehensibility, which may significantly enhance their performance in controllable image generation.

# 3 A NEW CONTROLLER FOR TEXT-TO-IMAGE GENERATION

In this section, we elucidate the theoretical analysis for extracting text encodings from decoder-only LLMs. Initially, we reveal that Transformer-based LLMs can be rephrased as diffusion model. Within this view, we pinpoint a specific timestep in the decoder component of an encoder-decoder LLM to deduce the encoder's text encoding distribution from its input and output. This deduction is then extended to decoder-only models, leading to the conclusion

that text encodings can be estimated from the outputs generated for sentences and words at each timestep.

## 3.1 Text-to-Image Diffusion Models

Text-to-image diffusion models typically employ an encoder to encode textual inputs $x$ with $d$-1 tokens as control conditions $c_{<d}$. Sequentially, those text encodings $c_{<d}$ are decoded for *image generation* through the diffusion model, *i.e.*, $p(z_{t-1}|z_t, c_{<d})$, where $z_t$ is the latent at timestep $t$, or for *text generation* via a text decoder, *i.e.*, $p(x_d|c_{<d})$, where $x_d$ is the $d$-th predicted token.

Typically, the text encoder utilized by diffusion models is derived from pre-trained models such as encoder-only or encoder-decoder LLMs. However, despite their impressive generative performance, decoder-only LLMs are not applicable to text-to-image generation. This is because these models directly generate tokens, making it infeasible to get text features $c$ directly.

## 3.2 LLMs as Diffusion Models

We revisit the transformer-based LLMs from a probabilistic perspective, to help to derive the formal modeling of text encodings for text-to-image generation. Considering that LLMs in a transformer architecture have a sequence of transformer blocks with the same structure, it is intuitive to model the forward process in a diffusion-like manner. Take an encoder-only LLM, CLIP, for example. Each input token is first fed into an embedding layer. For similarity, the output of the embedding layer for the $d$-th token is taken as the input, denoted as $x_d^T$. Then, it goes through $T$ transformer blocks that perform the causal attention masks in self-attention, which can be represented as $p_{\theta^t}(x_d^{t-1}|x_d^t, x_{<d}^t)$ for the $t$-th block parameterized $\theta_t$. This process is akin to the denoising process of DDPM with conditioning. So, the transformer-based LLMs can be viewed as diffusion models. Thus, we can leverage the dynamical properties and theoretical frameworks of diffusion models to analyze various structures of LLMs with a causal mask.

Moreover, the prediction of the model can be formulated as:

$$p_\theta(c_d|x_{\leq d}) = p(x_d^T) \prod_{t=1}^{T} p_{\theta^t}(x_d^{t-1}|x_d^t, x_{<d}^t) \qquad (1)$$

## 3.3 Text Encodings from Encoder-Decoder LLMs

For an encoder-decoder LLM, the encoder model processes contextual text, encoding it into a feature representation, *i.e.*, text encodings $c_{<d}$. Subsequently, the decoder model utilizes these text features to generate words with $p_{\theta^t}(x_d^{t-1}|x_d^t, c_{<d})$. Thus, each block in the decoder utilizes the same condition $c_{<d}$. Using the input $x_d^t$ and output $x_d^{t-1}$ of any block, the encoding $c_{<d}$ from the encoder is estimated through Bayes' theorem:

$$p(c_{<d}|x_d^{t-1}, x_d^t) = \frac{p(x_d^{t-1}, x_d^t|c_{<d})p(c_{<d})}{p(x_d^{t-1}, x_d^t)} \qquad (2)$$

## 3.4 Text Encodings from Decoder-only LLMs

For a decoder-only LLM, it is not directly available for textual features, *i.e.*, text encodings $c$ in encoder-decoder LLMs. Functioning as a generative model, it can be conceptualized as predicting the next

token based on conditions from the preceding tokens. Those contextual conditions are changing, not shared like encoder-decoder LLMs. Namely, when predicting the $d$-th word, the preceding $d$-1 words collectively serve as its contextual condition,

$$p_\theta(x_d|x_{<d}) = p(x_d^T) \prod_{t=1}^{T} p_{\theta^t}(x_d^{t-1}|x_d^t, x_{<d}^t) \quad (3)$$

Accordingly, given the input $x_d^t$ and output $x_d^{t-1}$ of transformer blocks, the estimation of $p(x_{<d}^t|x_d^{t-1}, x_d^t)$ can be derived as follows,

$$
\begin{aligned}
p_{\theta^t}(x_{<d}^t|x_d^{t-1}, x_d^t) &= \frac{p(x_d^{t-1}, x_d^t|x_{<d}^t)p(x_{<d}^t)}{p(x_d^{t-1}, x_d^t)} \\
&= \frac{p(x_d^{t-1}|x_{\le d}^t)p(x_d^t|x_{<d}^t)p(x_{<d}^t)}{p(x_d^{t-1}|x_d^t)p(x_d^t)} = \frac{p(x_d^{t-1}|x_{\le d}^t)p(x_{<d}^t|x_d^t)}{p(x_d^{t-1}|x_d^t)} \\
&\propto \underbrace{p(x_d^{t-1}|x_d^t, x_{<d}^t)}_{\text{block prediction of sentence}} \Big/ \underbrace{p(x_d^{t-1}|x_d^t)}_{\text{block prediction of single words}}
\end{aligned}
\quad (4)
$$

where $p(x_d^{t-1}|x_d^t, x_{<d}^t)$ is the generative LLM's prediction for $x_d^t$. Most existing LLMs employ a causal mask as the attention mask. Consequently, $p(x_d^{t-1}|x_d^t)$ can be obtained by feeding $x_d^t$ alone into the LLM, i.e., $p(x_d^{t-1}|x_d^t) = p(x_d^{t-1}|x_d^t, \emptyset)$.

However, it is still intractable to compute the text encodings $c$ from decoder-only LLMs. Notably, $x_{<d}^t$ is taken as the condition of the next token prediction, playing the similar role of $c_{<d}$ in encoder-decoder LLMs. Thus, there exists a $c_{<d}$ for decoder-only LLMs, which is the unbiased estimator of $x_{<d}$. Given that the decoder-only LLM can be viewed as diffusion model, we can estimate the score function of $p(c_{<d}|x_d^{t-1}, x_d^t)$ through $p_{\theta^t}(x_{<d}^t|x_d^{t-1}, x_d^t)$, thereby obtaining the text encoding $c_{<d}$. In accordance with Eqn. (4), the score function of $p_{\theta^t}(c_{<d}|x_d^{t-1}, x_d^t)$ can be approximated as follows:

$$
\begin{aligned}
\nabla_c \log p_{\theta^t}&(c_{<d}|x_d^t, x_d^{t-1}) \approx \\
&g(t)(\nabla_x \log p_{\theta^t}(x_d^{t-1}|x_d^t, x_{<d}^t) - \nabla_x \log p_{\theta^t}(x_d^{t-1}|x_d^t)),
\end{aligned}
\quad (5)
$$

where $g(t)$ is a scalar function that is dependent on the time step $t$. Furthermore, from Eqn. (1), by modeling an LLM as a diffusion process, the score function for $p(c_{<d}|x)$ can be approximated as:

$$
\begin{aligned}
\nabla_c \log p_{\theta^t}&(c_{<d}|x_d^t, x_d^{t-1}) \approx \\
&g(t)\big(\log p_{\theta^t}(x_d^t|x_{\le d}^{t+1}) - \log p_{\theta^t}(x_d^{t-1}|x_{\le d}^t)\big) \\
&-g(t)\big(\log p_{\theta^t}(x_d^t|x_d^{t+1}) - \log p_{\theta^t}(x_d^{t-1}|x_d^t)\big).
\end{aligned}
\quad (6)
$$

Taking into account the stochastic nature of generative language models during sampling, we can use this score function to perform Langevin dynamics sampling to obtain the final text encoding for image generation:

$$c_{<d}^{t-1} = c_{<d}^t + \nabla_c \log p_{\theta^t}(c_{<d}|x_d^t, x_d^{t+1}) + \sqrt{2h(t)}\epsilon_t, \quad (7)$$

where $h(t)$ is a learnable function, and $\epsilon_t \sim \mathcal{N}(0, I)$.

# 4 LLMDIFF ADAPTER

## 4.1 Decoder-only LLMs as Diffusion Controller

As discussed in Sec. 3, we can derive text encodings suitable for controlling diffusion image generation models from decoder-only

---

**Algorithm 1:** Text encoding from decoder-only LLMs

**Input:** Text input $x$ with length $D$, embedding layer $\omega$.

1   $c = \omega(x) \sim p(c^T)$ ; // Initial the text diffusion process.

    // denoise steps.

2   **for** $t = T$ **to** 1 **do**

3     **for** $d = 1$ **to** $D$ **do**

      // estimate $\nabla_x \log p_{\theta^t}(x_d^{t-1}|x_d^t, c_{<d})$.

4       $s_{sentence} \leftarrow g(t)S_{\theta^t}(x_d^{t-1}, x_d^t, x_{<d}^t)$;

      // estimate $\nabla \log p_{\theta^t}(x_d^{t-1}|x_d^t)$.

5       $s_{word} \leftarrow g(t)S_{\theta^t}(x_d^{t-1}, x_d^t)$;

6       $\nabla \log p_{\theta^t}(c_{<d}|x_d^t, x_d^{t-1}) \leftarrow s_{sentence} - s_{word}$;

7     **end**

8     $\epsilon_t \sim \mathcal{N}(0, 1)$;

9     $c \leftarrow c + \nabla \log p_{\theta^t}(c_{<d}|x_d^t, x_d^{t-1}) + \sqrt{2h(t)}\epsilon_t$;

10 **end**

 **Output:** $c$

---

LLMs utilizing Langevin dynamics:

$$c_{<d} = c_{<d}^T + \sum_{T-1}^{t=0}\left(\nabla_c \log p_{\theta^t}(c_{<d}|x_d^t, x_d^{t+1}) + \sqrt{2h(t)}\epsilon_t\right). \quad (8)$$

Leveraging the residual structure of existing transformer blocks and by combining Eqns. (5) and (6), we can transform these transformer blocks to derive the model for predicting scores: $S_{\theta^t}(x_d^{t-1}, x_{\le d}^t) \approx \nabla_x \log p(x_d^{t-1}|x_d^t, x_{<d}^t)$, $S_{\theta^t}(x_d^{t-1}, x_d^t) \approx \nabla_x \log p(x_d^{t-1}|x_d^t)$. Accordingly, the estimation of $c$ is implemented by Algorithm 1. Based on this text encoding, we can construct an adapter to integrate decoder-only LLMs into existing diffusion models. In contrast to the primary practice of merely employing LLMs for text optimization and subsequently encoding texts via a text encoder with inherent performance limitations, text encodings derived from LLMs to control the generation process of diffusion models can be a superior alternative for diffusion model training from scratch or adaption in a pre-trained diffusion model. In the following, we will elaborate an effective adaptor in a pre-trained diffusion model for image generation.

## 4.2 LLMDiff Adapter: Bridging Decoder-Only LLMs and Pre-trained Diffusion Models

To leverage the pre-trained knowledge of existing diffusion models more effectively, we propose an LLMDiff Adapter incorporating text encoding from generative decoder-only LLMs into a pre-trained text-to-image diffusion model, as illustrated in Fig. 3. The original cross-attention module is aligned with the preceding text encoder, and it is what actually imposes a bottleneck on the comprehension of user prompts. However, it still holds a wealth of knowledge and insights for text-to-image generation, learned during the pre-training phase. Therefore, we keep the original cross-attention module intact and align it with the encoding derived from LLMs through linear layers. This enables effective utilization of the knowledge of large-scale pre-trained models, preserving basic generation capabilities.

Simultaneously, an additional cross-attention module is introduced to learn how to better generate images based on the text

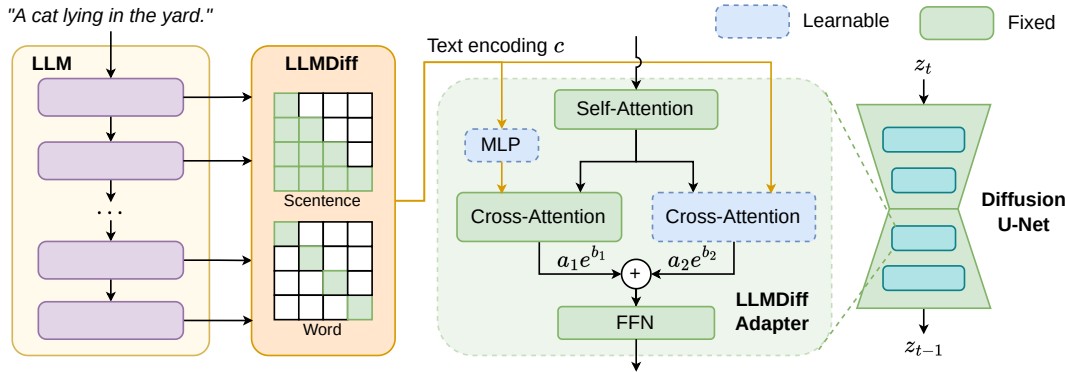

**Figure 3: Our LLMDiff-Adapter framework, wherein the parameters of both the LLM and the diffusion U-Net (including the original cross-attention module) are frozen during training. The newly added cross-attention module employs two adaptive-weight parameters to incorporate with the original one, which is dynamically adjusted during training.**

encoding derived from LLMs. The outputs of these two modules are combined through a set of learnable weight factors: $a_1$, $a_2$, $b_1$, and $b_2$, and the overall computation can be formulated as follows:

$$f = attn\big(\hat{\tau}_q(q), \hat{\tau}_k(\phi(\boldsymbol{c})), \hat{\tau}_v(\phi(\boldsymbol{c}))\big)a_1 e^{b_1} + \\ attn\big(\tau_q(q), \tau_k(\boldsymbol{c}), \tau_v(\boldsymbol{c})\big)a_2 e^{b_2}, \quad (9)$$

where $\hat{\tau}$ is the linear layer of the original cross-attention module, $\tau$ is that in additional cross-attention module, and $\phi$ is the linear layer to align the LLMs with the original cross-attention module. For training stability, the initial values of $a_1$ and $b_1$ are set to 1 and 0, respectively, while $a_2$ and $b_2$ start at 0.

During the model learning, the newly added cross-attention module gradually refines the outputs, effectively adapting the knowledge of generative LLMs to the diffusion model. Our Adapter is trained with the MSE loss for diffusion models:

$$\mathcal{L} = \|\epsilon_\theta(z_t, \boldsymbol{c}) - \epsilon\|^2, \quad (10)$$

where $\epsilon_\theta$ is the diffusion U-Net, $z_t$ is the latent feature map at timestep $t$, and $\epsilon \sim \mathcal{N}(0, I)$.

## 5 EXPERIMENTS

### 5.1 Experimental Settings

**Dataset.** We utilized a subset of data collected from GRIT [23] and midjourney-v5-202304-clean [30]. Simple filtering was applied to the image resolution and texts, with the total amount of data used for training approximating 1 million. To ensure a fair comparison, our model was trained alongside existing text-encoder-based models (including SD1.5 [27], SDXL [24], and T5-based SD1.5 models) using the same dataset and similar Adapters.

**Base models.** Our experiments are conducted based on pre-trained Stable Diffusion (SD) 1.5 model [27], utilizing two Large Language Models (LLMs), Phi1.5 [17] and Vicuna1.5-7B [6]. The number of parameters of Phi1.5 is close to that of the text encoders of CLIP and T5, thereby ensuring a fair comparison of performance.

**Implementation details.** Our LLMDiff Adapter has approximately 45M parameters. We utilize AdamW optimizer with a learning rate of 1e-5 for the Adapter training. The size of input images is 512x512, in conjunction with the Aspect Ratio Bucket, which automatically

**Table 1: Quantitative analysis of our LLMDiff Adapter compared with existing methods.**

| Method | SigLIP Score ↑ | Quality↑ | Complexity↑ | Beauty↑ |
|---|---|---|---|---|
| SD1.5 | 4.6 | 74.1 | 23.2 | 88.9 |
| SDXL | 6.2 | 76.5 | 23.9 | 90.6 |
| SD1.5 +(T5-XL) | 7.4 | 74.9 | 22.5 | 90.9 |
| Ours (phi1.5) | 5.8 | 76.3 | **24.9** | 91.0 |
| Ours (Vicuna-7B) | **8.5** | **78.6** | 24.7 | **92.9** |

groups images of different aspect ratios into different batches and seeks to avoid image cropping as much as possible. The weighted coefficients of the two cross attentions are initialized as follows: $a_1 = 1$, $b_1 = 0$, $a_2 = 0.1$, $b_2 = 0$. LLM itself does not require fine-tuning and we use a batch size of 256 for training on 8 NVIDIA A100 GPUs with 40GB VRAM.

**Metrics.** We assess the models from three dimensions. (1) For *controllability*, we evaluate the degree of matching between the generated images and the given text via the CLIP Score. However, since the SD model itself is based on CLIP, for fairness, we employ the SigLIP-L-384 [32] model to calculate the SigLIP Score:

$$Score(I, L) = 100 \times sigmoid(\alpha cos(f_{img}(I), f_{text}(L)) + \beta), \quad (11)$$

where $I$ is the input image, $L$ is the input text, $f_{img}$ is the image encoder, and $f_{text}$ is the text encoder. $\alpha$ and $\beta$ are the learned parameters from SigLIP model. (2) For *image quality*, we utilize CLIP-IQA [29] to evaluate the quality of the images from the aspects of image details and overall image quality. (3) As for the *logicality* of images, we employ the user study. For each model, we construct 15 prompts from multiple perspectives, including action logic, color matching, and the number of objects, etc. Each prompt generates 10 images, and human evaluators judge whether the core logic of these prompts is reflected in the images.

### 5.2 Quantitative Analysis

For a quantitative analysis of our method, we use the SigLIP Score to evaluate how well the generated images match the given text. Furthermore, we use CLIP-IQA to analyze the image's Quality,

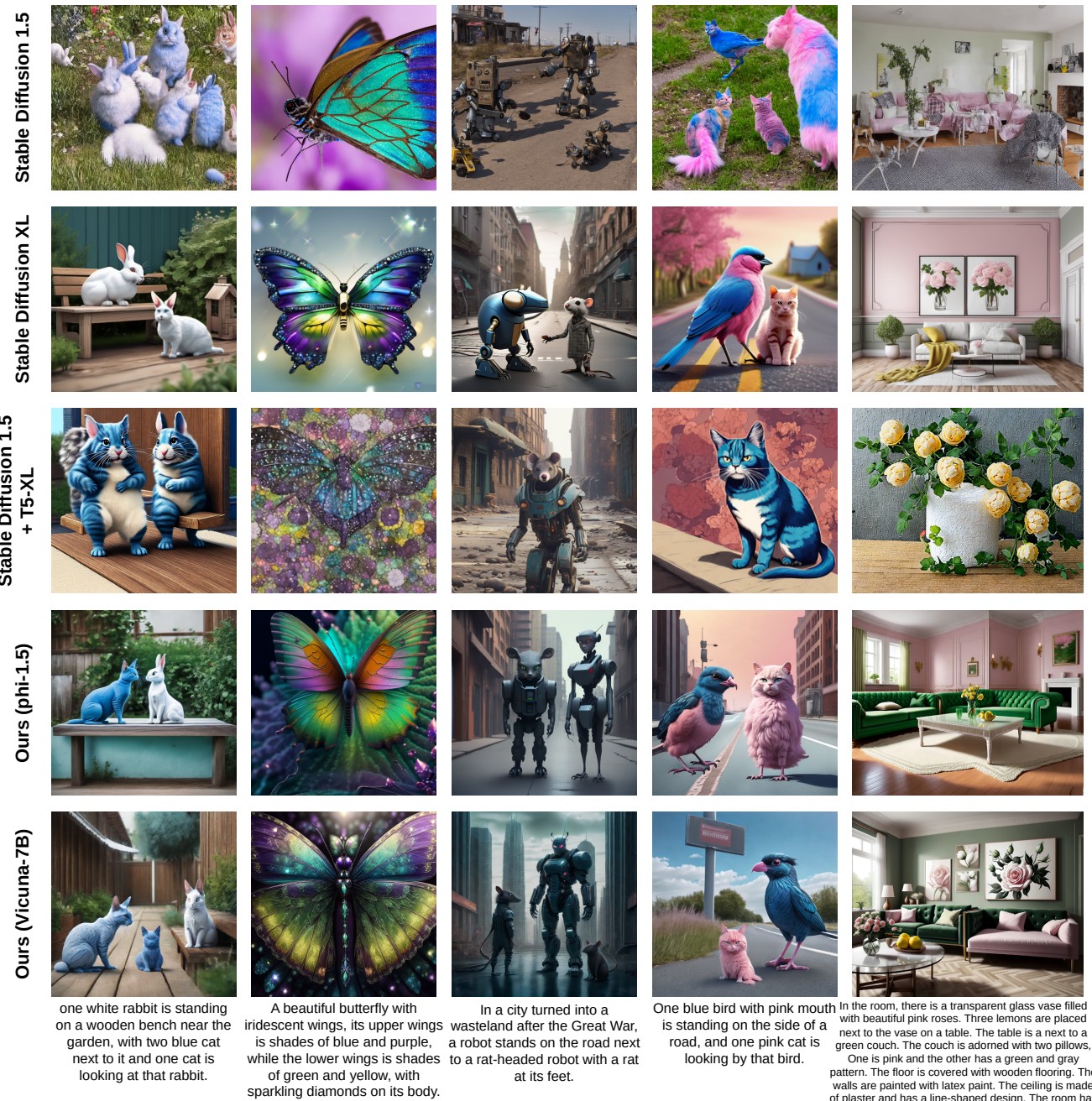

**Figure 4: In comparison with existing approaches, LLMDiff exhibits superior capabilities in both language comprehension and action understanding. Furthermore, it is proficient in generating images with high-quality details.**

Complexity, and Beauty, thereby assessing whether the overall quality of the generated image is better.

In Tab. 1, our proposed method based on Vicuna-7B achieved a SigLIP Score of 8.5, which is 31% higher than the best existing SDXL model of 6.2. Meanwhile, the model based on phi-1.5 has a 26% improvement compared to the SD1.5 used as our baseline, approaching the level of SDXL. These results suggest that our LLMDiff Adapter

can effectively combine the existing LLM and Diffusion models. Allowing the powerful text comprehension capabilities of the LLM to be utilized in the text-to-image diffusion model, thereby generating images with sufficient controllability. The more powerful the LLM, the stronger the controllability it brings.

Regarding the quality of the generated images in Tab. 1, our method surpasses existing methods in multiple aspects such as

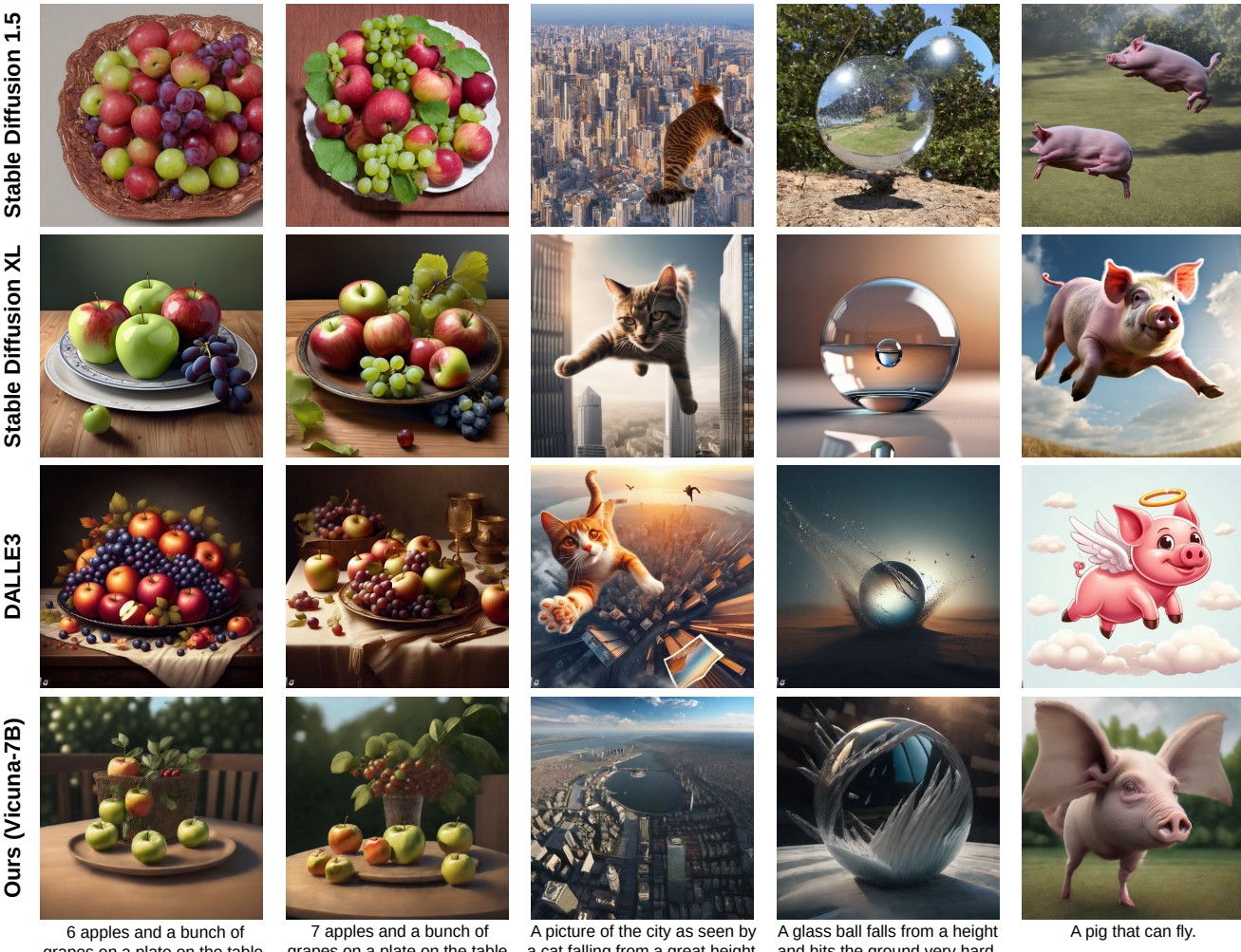

**Stable Diffusion 1.5** — **Stable Diffusion XL** — **DALLE3** — **Ours (Vicuna-7B)**

6 apples and a bunch of grapes on a plate on the table.

7 apples and a bunch of grapes on a plate on the table.

A picture of the city as seen by a cat falling from a great height.

A glass ball falls from a height and hits the ground very hard.

A pig that can fly.

**Figure 5: Model evaluation on the capability of causal and logical reasoning for text-to-image generation.**

overall image quality, complexity of details, and aesthetic appeal of the image. The Quality score reached 78.6, improving by 2.7% compared to the currently best SDXL model. In terms of the complexity of the generated image features, the Complexity score reached 24.7, with an improvement of 3.3% compared to SDXL. The aesthetic appeal score of the image also reached 92.9, surpassing the existing SDXL by 2.5%. By controlling the generation process of the Diffusion model through the LLM model, we can not only improve the alignment between the image and the text, but also enhance the quality, detail, and aesthetic appeal of the generated images.

## 5.3 Qualitative Evaluation

Our model is evaluated qualitatively from several aspects, encompassing its capacity to comprehend actions, entity relationships, spatial structures, and complex descriptions. As depicted in Fig. 4, our approach, powered by the robust semantic comprehension capabilities of LLMs, yields a more precise and controllable outcome in the portrayal of multiple entities and their inter-relationships. For instance, in the first column of Fig. 4, our method accurately

generates an image of a white rabbit seated on a wooden bench with two blue cats next to it, demonstrating a refined understanding of entity quantity and color correspondence. Furthermore, it exhibits a precise comprehension of the action wherein only one cat is looking at that white rabbit. In contrast, existing methods struggle to understand inter-entity relationships, like actions.

The third column further underscores our method's ability to accurately generate distinct entities as specified in the description, without any feature confusion. Instead, existing models tend to prioritize keyword comprehension and struggle to understand the holistic context of the sentence. We described three entities: a robot, a robot with a rat head, and a rat. A keyword-based approach risks overlooking some entities due to keyword overlap. Existing models leveraging text encoders like those of CLIP or T5 fail to accurately understand these inter-entity relationships and fail to generate images precisely when entity descriptions have keyword overlaps.

When it comes to generating complex scenes from extensive text descriptions, prevailing methods struggle to accurately comprehend the provided long text, resulting in generated scenes that

often diverge from the description, omitting numerous features. Our method, instead, integrates the power of LLMs into the diffusion model, exploiting LLMs' powerful long-text comprehension capabilities to precisely delineate each part of the scene, as well as the inter-entity relationships. For instance, as indicated in the last column of Fig. 4, our method can accurately render the complex indoor scene described in the extensive text, encompassing the pink rose, three lemons, a sofa with pillows, and the interior decoration. In contrast, existing methods fall short in accurately generating these complex features, typically producing simpler room scenes with a dearth of detailed features.

Furthermore, our method exhibits promising abilities in generating various meaningful detailed features. Our approach is based on the SD1.5 model, but when combined with the remarkable comprehension capabilities of LLM, our LLMDiff Adapter can significantly enhance the overall texture and detail quality of the generated images. The features generated by SD1.5 are typically fragmented, encompassing indistinguishable local features that are challenging to understand. In contrast, our method is capable of generating features with better coherence and more meaningful local features, particularly in complex scenes described by long texts.

## 5.4 Analysis of Reasoning Ability

Existing text-to-image generation models tend to produce visually highly similar image details with the given texts. They are struggle to generate image details that texts does not explicitly indicate, but is necessary for commonsense or reasoning. In our experiments, this is also taken into consideration for model evaluation, in Fig. 5.

In the first two columns of Fig. 5, our model can accurately understand the number of entities in the description, which is a great challenge for current diffusion models based on text encoders. With the help of LLM's understanding of quantifiers and entity relationships, we can enable the diffusion model to accurately generate the number of entities given in the description text.

In the third column of Fig. 5, the model is tasked with generating an image from the perspective of what a cat would see when it falls from a great height. The subject is not the cat itself, but rather the scene as viewed by the cat. This requires the model have enough capacity of logical reasoning. A model that primarily relies on keywords to interpret sentences can easily produce an image of a cat falling from a great height. Conversely, our method, grounded in LLMs, comprehends user intent well and generates the scene as perceived by the cat, not an image of the falling cat, despite the presence of the keyword "cat" in our description.

The fourth column illustrates how our model effectively leverages the LLM's capability to infer the physical rules. The task is to generate an image of a glass ball falling from a great height. In accordance with physical rules, the glass ball will inevitably shatter upon impact, making a lot of internal cracks that should propagate upwards from the point of impact, surrounded by fragments produced by the shattering of the glass ball. Existing methods lack this reasoning ability: the glass balls they generate remain intact, contradicting objective physical rules. Our method, instead, is able to draw these inferences accurately, generating the features implied in the description, including a shattered glass ball and a glass ball

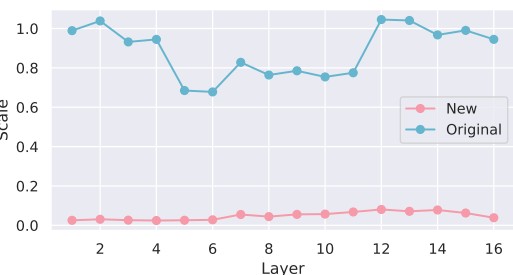

**Figure 6: The scale factor of newly added attentions and the original attentions in each cross-attention module of U-Net.**

that begins to fracture from the bottom due to the impact. These are what text-encoder-based diffusion models fail to comprehend.

The last column reveals our model's capacity to utilize LLM's inherent imagination ability and understanding of functions. The goal is to create a pig that can fly, focusing on its inherent ability to fly rather than its state or actions. It is expected that the generated image should depict an animal with a pig's primary features but a body structure adapted for flight. Existing models all generate a pig flying in the sky, and particularly, SD models simply draw a pig floating in the sky, without any imagination about the function of flying. Our model, instead, infers the user's intention from the text and conceptualizes the ability to fly based on LLM's knowledge base. It borrows structure characteristics from common flying animals, like birds, and integrates them into the pig. The pig's ears evolve into larger structures resembling wings, and it reduces to two feet and a smaller size, which are typical bird traits. Our model thus imagines a pig with flying capabilities rooted in real-world logic and pig features, rather than forcibly attaching wings.

## 5.5 Analysis of Scaling Factors

According to the results in Fig. 6, which shows the weight distribution across various layers in the new and original cross-attentions, a substantial portion of the original knowledge within the model is preserved. The preservation is less in layers with a higher number of parameters, while layers at both ends, which have fewer parameters, retain more. The newly added rectification module primarily operates in the decoder part of the U-Net.

## 6 CONCLUSION AND LIMITATIONS

We viewed generative LLMs with a transformer-based decoder-only structure as a diffusion model, thereby we can sample implicit text encodings for image generation. We have designed the LLMDiff Adapter to incorporate these encodings into a text-to-image diffusion model, enhancing the model's controllability and reasoning abilities pertaining to commonsense, logic, and physics. The generated images are more realistic, with improved detail and quality. Moreover, our method outperforms existing text-encoder-based methods in various quantitative metrics.

**Limitations.** Our method requires the output from each Transformer block of the LLM, and thereby is incompatible with closed-source models like GPT-4 and Claude 2.

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
