# OpenReview forum: "Decoder-Only LLMs are Better Controllers for Diffusion Models"
_acmmm.org/ACMMM/2024/Conference — MM2024 Poster_

### Official Review · Reviewer_aUz5 · 2024-05-16

**Rating:** 4
**Confidence:** 3

**Summary:**

This paper proposes an innovative framework for controlling text-to-image generation using Large Language Models (LLM). The framework connects LLMs and Diffusion Models through the LLMdiff Adapter. The paper provides theoretical analysis as well as qualitative and quantitative experiments to validate the effectiveness of the proposed method.

**Strengths:**

1. The paper is well-written and well-structured.
2. This paper uses LLM as a text controller to guide the Diffusion Model that does not need further training in the UNet and LLM, which is somewhat innovative.
3. The experimental results show that LLM can outperform the baseline, such as SD1.5 and SD-XL. In some cases, this method outperforms DALL-E 3.

**Limitations:**

1. Figure 2 requires further clarification. The current version does not clearly demonstrate the advantages of decoder-only LLMs over encoder-decoder structures.
2. Some experimental settings are not clear. Which dataset are the quantitative experiments conducted on?
3. The experiments are limited. Lack of comparison on current popular decoder-only architectures, e.g., Llama. Additionally, Pixel-Art α, which also uses T5-XL as a text encoder, does not compare to your method. The text-to-image generation results have not been tested on the T2I-CompBench benchmark.

**Suitability:**

3

---

### Official Review · Reviewer_3fUr · 2024-05-25

**Rating:** 5
**Confidence:** 4

**Summary:**

The paper attempts solving the issue of limited semantic understanding in text-to-image generation models. The authors proposed an adapter to incorporate the knowledge from LLMs into the cross attention conditioning blocks of diffusion models - with the goal being to leverage the rich knowledge these large scale trained LLMs have.

**Strengths:**

Paper is well written and insightful. The direction of approach is interesting, and is different from the existing LLM directions where the utility of LLM has been limited to coming up with just more detailed input prompts. The evaluation protocols to evaluate multiple aspects (e.g. entity relationship, actions) is extensively designed.

**Limitations:**

I am overall positive about the novelty, technical depth and quality of the work.
The only aspect is lack of comparison with baselines - there are many works in the direction of prompt optimization with or without using LLMs. Would like to see how the proposed method compares with them.

**Suitability:**

3

---

### Official Review · Reviewer_j6Xe · 2024-05-27

**Rating:** 4
**Confidence:** 2

**Summary:**

In this paper, the authors propose an approach to text-to-image diffusion models by integrating large language models (LLMs) with a decoder-only architecture. The proposed method aims to enhance semantic understanding and generate higher quality images from text prompts. Traditional text-to-image models use an encoder structure which often struggles with understanding complex, detailed prompts and lacks in generating high-fidelity images. The authors introduce an "LLMDiff-Adapter" that allows seamless integration of decoder-only LLMs with diffusion models, improving image generation quality, reliability, and text comprehension.

**Strengths:**

- The authors provide detailed theoretical analysis in Section 3.4.
- It seems that images generated using the proposed method are superior in quality and detail compared to those produced by state-of-the-art models like DALL-E 3 and Stable Diffusion.

**Limitations:**

- The approach heavily relies on the capabilities of large pre-trained language models, which increases the complexity and requires significant computational resources.
- It would be great if the authors could include and analyse some failure cases.
- There is a lack of discussions on how the proposed method compares to other possible integrations of LLMs with diffusion models that might use different architectural or training strategies.
- There are too few details regarding the user study of this paper.

**Suitability:**

3

---

### Meta-Review · Senior_Area_Chairs · 2024-07-10

**Recommendation:** Accept (Poster)
**Confidence:** 5

**Metareview:**

This paper proposes a new approach for text-to-image generation aiming to improve the semantic understanding capabilities of the model and thus create a better image. It is clearly a paper that fits ACM Multimedia (it has interesting techniques as well as a user study). All reviewers indicate several positive aspects of the method but also indicate that the method could have done better in comparing to existing baselines. All in all the scores come to 2 borderline accepts and one weak accept. Overall I feel that the paper can be accepted.